# Antibody Profile, Gene Expression and Serum Cytokines in At-Risk Infants before the Onset of Celiac Disease

**DOI:** 10.3390/ijms24076836

**Published:** 2023-04-06

**Authors:** Renata Auricchio, Martina Galatola, Donatella Cielo, Roberta Rotondo, Fortunata Carbone, Roberta Mandile, Martina Carpinelli, Serena Vitale, Giuseppe Matarese, Carmen Gianfrani, Riccardo Troncone, Salvatore Auricchio, Luigi Greco

**Affiliations:** 1Department of Translational Medical Science, University Federico II, 80131 Naples, Italy; martinagalatola@hotmail.it (M.G.); dona_cielo@yahoo.it (D.C.); robertarot.97@gmail.com (R.R.); rmandile91@gmail.com (R.M.); martinacarpinelli@gmail.com (M.C.); troncone@unina.it (R.T.); 2European Laboratory for Food Induced Diseases, University Federico II, 80131 Naples, Italy; carmen.gianfrani@ibbc.cnr.it (C.G.); salauric@unina.it (S.A.); ydongre@unina.it (L.G.); 3Laboratory of Immunology, Institute for Experimental Endocrinology and Oncology, National Research Council of Italy (IEOS-CNR), c/o Department of Molecular Medicine and Medical Biotechnology, University Federico II, 80131 Naples, Italy; fortunata.carbone@ieos.cnr.it (F.C.); giuseppe.matarese@unina.it (G.M.); 4Neuroimmunology Unit, IRCCS Fondazione Santa Lucia, 00179 Rome, Italy; 5Institute of Biochemistry and Cell Biology, National Research Council of Italy (IBBC-CNR), 80131 Naples, Italy; serena.vitale@ibbc.cnr.it

**Keywords:** celiac disease, prospective cohorts, infants at risk for celiac disease, anti-gliadin antibodies, anti-tissue transglutaminase antibodies, serum cytokines and gene expression, tolerance

## Abstract

Immunological events that precede the development of villous atrophy in celiac disease (CeD) are still not completely understood. We aimed to explore CeD-associated antibody production (anti-native gliadin (AGA), anti-deamidated gliadin (DGP) and anti-tissue transglutaminase (anti-tTG)) in infants at genetic risk for CeD from the Italian cohorts of the PREVENT-CD and Neocel projects, as well as the relationship between antibody production and systemic inflammation. HLA DQ2 and/or DQ8 infants from families with a CeD case were followed from birth. Out of 220 at-risk children, 182 had not developed CeD by 6 years of age (CTRLs), and 38 developed celiac disease (CeD). The profiles of serum cytokines (INFγ, IL1β, IL2, IL4, IL6, IL10, IL12p70, IL17A and TNFα) and the expression of selected genes (FoxP3, IL10, TGFβ, INFγ, IL4 and IL2) were evaluated in 46 children (20 CeD and 26 CTRLs). Among the 182 healthy CTRLs, 28 (15.3%) produced high levels of AGA-IgA (AGA+CTRLs), and none developed anti-tTG-IgA or DGP-IgA, compared to 2/38 (5.3%) CeD infants (Chi Sq. 5.97, *p* = 0.0014). AGAs appeared earlier in CTRLs than in those who developed CeD (19 vs. 28 months). Additionally, the production of AGAs in CeD overlapped with the production of DGP and anti-tTG. In addition, gene expression as well as serum cytokine levels discriminated children who developed CeD from CTRLs. In conclusion, these findings suggest that the early and isolated production of AGA-IgA antibodies is a CeD-tolerogenic marker and that changes in gene expression and cytokine patterns precede the appearance of anti-tTG antibodies.

## 1. Introduction

Celiac disease (CeD) is characterized by chronic intestinal inflammation caused by an abnormal immune response to prolamins found in wheat and other cereals [1]. Prolamins not tolerated by CeD patients contain specific sequences rich in proline and glutamine amino acids that are resistant to human gastrointestinal proteases and therefore remain partially digested in the intestinal tract [2,3,4,5]. At the site of the intestinal mucosa, the human tissue transglutaminase enzyme (tTG) deamidates specific glutamine (Q) residues, which facilitates the presentation of gluten peptides to intestinal pro-inflammatory T cells [6]. At the same time, antibodies against tTG are produced, representing the diagnostic hallmark of the disease [7]. 

The genetic profile [8,9,10] and a set of environmental factors contribute to the multiplicative risk of developing the disease [9,10,11,12,13]. In fact, prenatal events, early feeding patterns [11,12,13,14], viral infections [15,16] and other unknown factors may contribute to an increased risk of CeD. 

Recently, in the German and Hungarian cohorts of the PREVENT-CD project, the profile of antibody development was explored [17]. In their recently published study, Diós et al. evaluated the progression from the early recognition of the native gliadin peptide, resulting in the production of anti-gliadin antibodies (AGAs), to the production of the anti-deamidated gliadin antibody (DGP), and finally to the development of anti-tTG autoantibodies. The authors suggested that the simple recognition of the antigen and the exclusive production of AGAs are not predictive of disease development [17]. However, the description of what happens between the loss of tolerance to gluten and the beginning of intestinal damage is still unclear. The humoral response to gliadin peptides, the deamidation of QXP-sequence-containing sequences [18] and the resulting formation of the complex with tTG [19], a prerequisite for the production of anti-tTG autoantibodies [20], are likely to be key to the development of the full-blown disease. The early production of AGAs could allow for distinguishing children who develop tolerance from those who progress to an abnormal immune response to gluten.

In order to investigate the role of AGA production in the development of tolerance after early dietary gluten exposure, here, we explored the relationship between the early production of antibodies with the development of villous atrophy in a longitudinal cohort of children at risk for CeD. In addition, we evaluated the levels of serum cytokines and the expression of a set of pro-inflammatory genes in relation to the production of antibodies. 

## 2. Results

### 2.1. Antibody Production in At-Risk Children Who Develop CeD

Thirty-eight genetically at-risk infants who developed CeD by 6 years of age frequently started to produce anti-tTG antibodies after the second year of life, with a peak at a median age of 42 months (Appendix A). The peaks of AGA and anti-tTG antibodies in CeD, which were estimated by the longitudinal profile of each child, are shown in Table 1. The peak of AGAs occurred slightly before the production of anti-tTG: the first peak was observed around 28 months and the highest peak occurred around 36–42 months of age, when anti-tTG antibodies were also detected. Similarly, in the few cases where DGP antibodies were estimated, their profile overlapped with that of AGAs (Appendix A). Indeed, in CeD (8 children, 50 samples), both DGP-IgA and DGP-IgG correlated strongly with anti-tTG (Pearson’s r = 0.669 and r = 0.807, both *p* < 0.0001). DGP-IgA also correlated with AGA-IgA (r = 0.365, *p* = 0.009); similarly, DGP-IgG correlated with AGA-IgG (r = 0.626, *p* = 0.0001) (Appendix A). The antibody patterns of some representative cases are reported in Appendix A, panel A. The antibody profile showed a clear overlap among anti-tTG, AGA and DGP. These data suggest that in CeD children, the production of both AGA and DGP antibodies overlaps with the production of anti-tTG. Table 1 shows the timing of antibody production peaks detected in both healthy controls (AGA+CTRLs, second column) and CeD children (CeD, third column). In the fourth column, the timing of the peak of anti-tTG in CeD is displayed.

### 2.2. Antibody Production in At-Risk Children Who Do Not Develop CeD (Control Children)

From 6 to 24 months, 28/182 (15.3%) at-risk children who did not progress to CeD (CTRLs) produced AGA-IgA antibodies above the 95° centile of the normal range (10 U/mL) and were hence named AGA+CTRLs, while only 2/38 (5.3%) CeD children produced significant amounts of AGAs at the same age. The difference in the production of early AGAs between AGA+CTRLs and CeD was statistically significant (Chi Sq. 5.97, *p* = 0.0014). It is noted that 5/28 (17.8%) CTRLs, who produced significant amounts of AGAs as early as 6 months, had been exposed to 100 mg of gluten from 4 to 6 months, according to the PREVENT-CD study protocol [21]. By 12 months of age, six CTRLs produced AGAs, and nine became AGA-positive by 24 months. The second column in Table 1 shows the distribution of the peaks of AGA antibodies in AGA+CTRLs by age, with a cluster of AGA peaks in the first year of life. The median age at the peak of AGAs was 19.6 months, with a geometric mean of 16 months, earlier than CeD children. No detectable production of anti-tTG was observed in these children throughout the follow-up period. Some individual profiles of AGA+CTRLs for which DGP antibodies were detected are reported in Appendix A, panel B. The production of AGA IgA was not associated with the production of DGP, and no correlation was found between the production of AGAs. 

### 2.3. Serum Cytokines

The production of a panel of innate and adaptive cytokines was measured in the sera of the children in the various groups. In particular, we detected the serum concentrations of nine cytokines, INFγ, IL1β, IL2, IL4, IL6, IL10, IL12p70, IL17A and TNFα, in 83 serum samples from 20 CeD patients (41 samples) and 26 CTRLs (42 samples) at 4 months of age (mean 3.85 months, range 3–5) before gluten introduction and after the second year of life (mean 36.2 months, range 24–48). Children who later developed CeD were examined at a median age of 20 months, more than one year before the first production of anti-tTG; hence, this phase is named “PRE-CeD”. These PRE-CeD children showed increased levels of all cytokines compared to CRTLs, as was also the case in the same CeD infants in their active diagnostic phase (Figure 1 and Appendix A). AGA+CTRLs did not show big differences in their serum profiles of pro-inflammatory cytokines compared to CTRLs, who did not produce AGAs at any point in their lives (Appendix A).

### 2.4. Cytokine Gene Expression 

The expression of several genes involved in the immune response, such as FoxP3, IL10, TGFβ, INFγ, IL4 and IL2, was evaluated in peripheral blood mononuclear cells (PBMCs) isolated from a total of 67 blood samples. The expression of the selected genes is reported in Figure 2, in which each panel shows a comparison between two different groups of children for greater clarity. In detail, panel A shows the comparison of gene expression between PRE-CeD and CeD groups, panel B compares CTRLs and PRE-CeD, panel C compares CTRLs and CeD, and panel D compares CTRLs and AGA+CTRLs. PRE-CeD showed a significantly lower expression of the INFγ gene compared with CeD, as expected but a similar expression of the FoxP3, TGFβ, IL10, IL4 and IL2 genes (Figure 2, panel A). PRE-CeD cases, as well as active CeD cases, showed lower TGFβ and FoxP3 expression compared to CTRLs (Figure 2, panels B and C). AGA+CTRLs showed significantly increased INFγ and IL4 gene expression compared to CTRLs who did not produce AGAs and a lower expression of FoxP3, IL10 and TGFβ (Figure 2, panel D).

### 2.5. Multivariate Analysis to Predict Early Antibody Production and Later Development of CeD 

Since the analyzed biomarkers are correlated, we aimed to find the best profile for each study group in order to distinguish children who develop a tolerogenic antibody response from those who progress to the full-blown disease. As summarized in Table 2, early AGA production in CTRLs is best predicted by the combination of IL4, FoxP3 and INFγ gene expression and, in addition, by the presence of the serum cytokine IL17A, with the correct discrimination of 85% CTRLs from AGA+CTRLs. PRE-CeD children are well (93%) differentiated from healthy CTRLs, regardless of the serum antibody production, by the presence of serum cytokines (IL2, IL4, IL12, INFγ, IL17A and IL10), along with the expression of IL10, IL4 and INFγ. This difference is similar to that observed between children with active CeD and CTRLs, so the hypothesis that PRE-CeD children have an inflammatory profile activated before the production of specific antibodies (DGP and anti-tTG) and the development of flat mucosa is supported. All PRE-CeD children are discriminated from AGA+CTRLs by IL17A and IL10 cytokines together with INFγ, FoxP3 and IL4 gene expression. Most inflammatory biomarkers are equally increased in both PRE-CeD and CeD groups; nevertheless, 94% of PRE-CeD children are correctly differentiated from those with full-blown CeD by a model that includes INFγ and TGFβ gene expression together with TNFα and IL1β serum cytokines.

## 3. Discussion

Longitudinal studies offer the opportunity to observe the progression from health to disease in autoimmune disorders: the PREVENT-CD project allowed a cohort of HLA DQ2/DQ8-positive babies from families with a proband CeD case to be followed from birth to 6 years of age and older [22]. The outcome for these babies [22] (16% developed full-blown CeD with villous atrophy in the Italian cohort) was related to the genetic profile of the infant [23] and to a series of environmental factors contributing to the multiplicative risk of developing the disease [23]. The environment in which the genetically predisposed infant grows may be crucial for the disease outcome: prenatal events, as well as feeding patterns and infections in the first years of life, may modulate gene expression [24]. In this study, we prospectively monitored CeD-associated antibody production, i.e., AGA, DGP and anti-tTG antibodies, in infants at genetic risk for CeD and correlated the antibody production profiles with systemic cytokine responses.

One unanswered question in CeD pathogenesis is the timing of the production of anti-gliadin antibodies and how it is linked to the development of anti-tTG autoantibodies. It was suggested that some at-risk infants who do not develop the disease show a transient production of antibodies against non-deamidated gliadin peptides (AGAs) upon gluten introduction, which is not associated with the response to deamidated gliadin peptides (DGP) or to the production of anti-tTG [17]. This study demonstrated that AGAs may appear very early, with no relation to specific CeD antibodies, particularly to anti-tTG [17]. Indeed, in the German and Hungarian cohorts of the same PREVENT-CD study, gliadin-specific antibodies that rose early in life after stimulation by precocious exposure to a small amount of gluten preferentially recognize non-deamidated gliadin, while the anti-gliadin antibodies produced late in children who became CeD preferentially recognize deamidated forms. These diverse antibody specificities suggest an active role of transglutaminase in disease progression [18]. Similarly, in our study cohort, 28/182 (24.5%) genetically predisposed children who did not develop CeD before 6 years of age produced detectable AGA antibodies from 8 to 40 months (AGA+CTRLs). However, these antibodies were not associated with anti-tTG, nor with specific disease conditions, as reported in the personal logbook. Thus, our findings suggest that the lack of HLA-DQ restricted pathogenic gluten-specific T cells with a helper function in CeD-associated B-cell responses. Overall, our data on humoral response profiles confirm that the anti-native gliadin immune response is not predictive by itself of the progression to CeD, since it may be present independently of the activation of the pathogenic mechanism mediated by tissue transglutaminase. It is noteworthy that it has been hypothesized that the formed transamidated tTG–gluten complexes act as hapten–carrier systems, leading to the production of specific anti-tTG antibodies, in which gluten-reactive CD4+ T cells have a key helper role [20,21,22,23,24,25].

The group of AGA+CTRLs presented a significantly higher expression of INFγ and IL4 genes when compared to controls who did not produce antibodies (CTRLs), confirming a non-specific and transient inflammatory profile, although it did not progress to the development of CeD. In contrast, children who developed CeD over time produced all types of anti-gliadin/tTG antibodies (AGA, DGP and anti-tTG), with an early peak around 28 months and a larger peak around 36–42 months of age. As expected, CeD children, when compared to healthy CTRLs, had a significantly higher expression of INFγ genes together with a reduction in regulatory FoxP3 and TGFβ gene expression. Notably, non-CeD control children have a significant increase in IL4, a cytokine that has been recently described as a biomarker of “healthy”, undamaged gut mucosa [26,27].

When we looked at the serum levels of a large panel of cytokines, including INFγ, IL1β, IL2, IL4, IL6, IL10, IL12p70, IL17 and TNFα, we observed an overall increase in all analyzed markers in CeD children compared to CTRLs. Instead, children who developed CeD, when examined 12–24 months before they were first positive for anti-tTG, a phase named PRE-CeD, showed a cytokine production profile that did not differ from the values recorded during the later active phase of the full-blown disease, associated with a higher expression of TGFβ and a lower expression of INFγ. These findings suggest that in PRE-CeD children, the inflammatory pathway leading to the disease has already started. These PRE-CeD children showed clear signs of immune activation, a lower expression of FoxP3 and TGFβ compared to CTRLs and higher levels of all evaluated serum cytokines before the appearance of the disease. It is worth noting that in this same Prevent-CD cohort, in one case, gliadin-reactive T cells were detected in the intestine of anti-tTG seronegative children with anti-gliadin antibody positivity and normal villous architecture [28]. In children with villous atrophy, these CeD-relevant T cells are present in the duodenum and display more intense IFNγ production in response to deamidated gliadins, in line with findings from other cohorts of CeD children [28]. This scenario appears to be influenced by a cumulative effect of multiple risk elements (genetic predisposition, inflammation, anti-tissue transglutaminase activation and the deamidation of gliadin peptides) and the presentation of gliadin peptides to pro-inflammatory helper T cells, which stimulate plasma cells to produce anti-DGP and anti-tTG antibodies [29]. Furthermore, we demonstrated that both the full-blown (CeD) and pre-symptomatic (Pre-CeD) phases are associated with the lower expression of genes involved in the development of regulatory cells, such as FoxP3 and TGFβ. It is possible that the limited activation of tolerance within our cohort of children at genetic risk for CeD could be crucial to determining the onset of the disease. Furthermore, we showed that in subjects at CeD risk, markers of inflammation are present before gluten introduction and could enhance an existing inflammation state elicited by other pro-inflammatory triggers, such as common viral infections [15]. 

Although our findings might be limited by the relatively small sample size and restricted set of biomarkers, their strengths are the longitudinal setup and the examination of the overlap between serum cytokines and the expression of a selected set of immune-regulatory genes. We demonstrated that, in our cohort of infants at risk for CeD, the evaluation of CeD-associated antibody production and their relationship with systemic parameters of inflammation, mediators of both regulatory and inflammatory pathways, could be predictive of disease onset. Future studies should explore the global “exposome”, which drives a certain number of at-risk children toward CeD through a series of epigenetic manipulations in early life, as indicated by gene expression before exposure to the causative antigen. The profile of the antibody pattern [30,31,32,33], as well as a peculiar metabolomics profile [34], is the outcome of such early changes.

## 4. Materials and Methods

### 4.1. Patients and Study Design

The EU-funded PREVENT-CD project allowed a cohort of HLA DQ2/DQ8-positive newborns from families with a proband CeD case to be followed from birth [22]. An Italian cohort of 220 HLA DQ2/DQ8+ infants from families with a celiac case was followed from birth to 6 years of age. In particular, about 90% of the enrolled children came from the metropolitan area of Naples, while the remaining 10% were from other territorial areas of the Campania region. Children were followed at the Department of Translational Medical Sciences, Section of Pediatrics, Federico II, University of Naples. They were tested for total serum IgA and celiac-disease-specific antibodies (anti-tissue transglutaminase, anti-tTG; anti-gliadin, AGA): 182 children did not develop anti-tTG antibodies up to 6 years of age, and we classified them as controls (CTRLs). By 6 years of age, 38 subjects had developed anti-tTG antibodies and underwent a small bowel duodenal biopsy: 2 showed a normal mucosa (Marsh stage 0/1) and were classified as potential celiac patients, and 36 showed severe mucosal damage (Marsh stage 3c) and were classified as cases (CeD). The median age at diagnosis was 38.5 months (range 18–60, SD 14.1). In this Italian cohort, we considered 4 groups of children, whose features are summarized in Table 3: 154 children did not develop CeD and had negative serology (CTRLs), 28 did not develop CeD but produced anti-gliadin antibodies (AGA+CTRLs), 38 CeD children who later developed CeD and examined at least 12 months before the appearance of anti-tTG antibodies (Pre-CeD), and finally, the same 38 CeD children at the time of diagnosis (CeD). The PREVENT-CD study protocol was a randomized placebo-controlled trial aimed at preventing the development of the disease. Therefore, 66/220 were exposed to 100 mg of gluten from the 4th to the 6th months of life, and the rest received a placebo (21). The AGA+CTRLs were more frequently (13/28; 46.4%) exposed to gluten compared to CTRLs, who did not produce antibodies (39/154; 25.3%). For a small subset of CeD children (8 patients, 50 samples collected) and AGA+CTRLs (8 patients, 46 samples collected), IgA and IgG anti-deamidated gliadin (DGP) serum levels were available at different time points. 

### 4.2. Estimation of Antibodies Titers

Measurements of total IgA, anti-tTG-IgA, AGA-IgA, and DGP-IgA or DGP-IgG in the sera were performed using Varelisa Gliadin IgA and Celikey (Phadia GbH, Freiburg, Germany; cut-offs of 17, 6 and 6 U/mL, respectively). Since the antibodies did not show a normal distribution, the decimal log was applied in order to normalize their distribution. Geometric means (means of log-transformed values) with their corresponding 95% confidence intervals are shown. In order to identify the children who produced significant amounts of AGA antibodies, we estimated the threshold of the 95% percentile of the distribution of AGAs in controls and calculated a value of 10 units/mL. The age at the peak of antibody production in each child was determined through the direct inspection of their individual profiles over time. Anti-DGP antibodies were analyzed in the first period of the PREVENT-CD study, but the test was soon abandoned, since it appeared to coincide with the production of anti-tTG antibodies. 

### 4.3. RNA Extraction and Gene Expression 

The expression of a set of genes involved in the immune response (FoxP3, IL10, TGFβ, INFγ, IL4 and IL2) in 67 peripheral blood mononuclear cell (PBMC) samples was estimated after RNA extraction, as described previously [35]. Gene expression was analyzed according to Minimum Information for Publication of Quantitative Real-Time PCR Experiments (MIQE) guidelines (http://www.gene-quantification.de/miqe-bustin-et-al-clin-chem-2009.pdf accessed on 15 March 2019). 

### 4.4. Inflammatory Cytokine Measurements 

We analyzed a set of 9 cytokines in 83 serum samples from 20 CED patients (41 samples) and 26 CTRLs (42 samples) at 4 months of age (mean 3.85, range 3–5) before gluten introduction and after the second year of life (mean 36.2 months, range 24–48). INFγ, IL1β, IL2, IL4, IL6, IL10, IL12p70, IL17A and TNFα were detected using High Sensitivity 9-Plex Human ProcartaPlex™ with Luminex 200™ (Luminex Corp., Austin, TX, USA), and XPONENT was used for data analysis. The detection limit ranged from 0.02 ng/mL to 38 pg/mL. According to the manufacturer’s specifications, the intra-assay and inter-assay coefficients of variation were <10% and <20%, respectively. The raw values of cytokine levels were transformed into the decimal log because of their skewed distributions.

### 4.5. Sample Size

To compare the AGA+CTRL group to CTRLs who did not produce antibodies, we set the following parameters: 25% of CeD-predisposed infants might produce AGA antibodies, and the CeD group could be unbalanced versus the other group (CTRLs) with an RR of 5, with a first-degree error of 0.05 and a second-degree error (power) of 0.80. In each group, the required sample size was 27.

### 4.6. Statistics

The distributions of raw values of antibody levels and serum cytokines were not compatible with normality; hence, decimal log transformation was performed for these values in order to normalize their distributions (Skewness and Kurtosis < 1). Geometric means (antilog of log10-transformed values) with 95% confidence intervals are shown. Pearson correlation coefficients were applied to evaluate relations between variables. The Rank-Sum test was used to compare gene expression values. Student’s t test was used to compare means of log10 serum cytokines, which showed a normal distribution after the log transformation. To deal with the bias of multiple comparisons, a multivariate canonical stepwise discriminant analysis was used to classify the different groups. Due to the relatively small size of each group, the multivariate model was only used to explore how each marker included in a hypothesis-free model contributes to distinguishing between groups. Wilk’s lambda estimates the capacity of each variable to discriminate between two groups and ranges from 1 (complete overlap between groups) and 0 (complete separation). The contribution of each variable to the discriminant model is cumulatively added to Wilk’s lambda. Based on the discriminant equation obtained by using the selected variables, the classification of each individual in one of the groups is obtained, and the % of correct classifications is computed.

## 5. Conclusions

In conclusion, we demonstrated the early production of gliadin IgA antibodies (AGAs) in a subset of infants at risk for CeD, but this is not predictive of the disease. On the contrary, the early production of AGAs seems to protect them. In contrast, children who later develop CeD show the simultaneous production of AGA, anti-DGP and anti-tTG antibodies. Furthermore, in those who develop the disease, changes in gene expression and cytokine patterns precede the appearance of anti-tTG antibodies. 

## Figures and Tables

**Figure 1 ijms-24-06836-f001:**
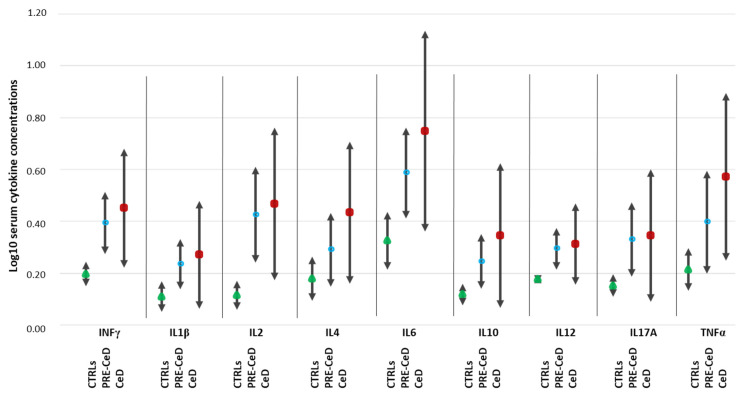
Serum cytokines in controls, celiacs before the onset of disease and active celiacs at time of diagnosis. The serum cytokine concentrations were detected in 83 serum samples from 20 celiac patients (41 samples) before (PRE-CeD) and after the diagnosis (CeD) and 26 controls (CTRLs, 42 samples) by multiplex protein quantitation using the Luminex instrument platform. The raw values of cytokine levels were transformed by the decimal logarithm. Geometric means of log10-transformed values (▪) are shown with 95% confidence intervals (↨).

**Figure 2 ijms-24-06836-f002:**
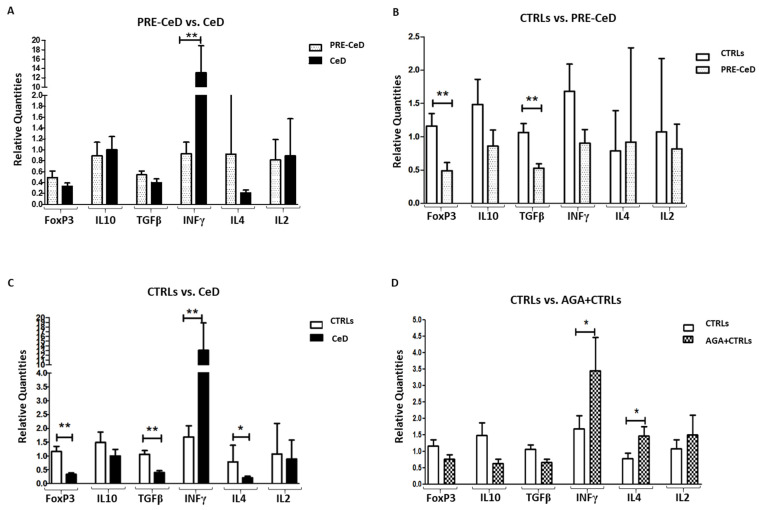
Relative Quantities (RQs) of the expression of selected cytokine genes. Gene expression of FoxP3, IL10, TGFβ, INFγ, IL4 and IL2 was observed in 67 blood samples and analyzed according to Minimum Information for Publication of Quantitative Real-Time PCR Experiments (MIQE) guidelines. (**A**) Gene expression comparison between patients before the diagnosis (PRE-CeD, dotted histograms) versus active celiacs (CeD, black histograms), (**B**) PRE-CeD (dotted histograms) vs. controls (CTRLs, white histograms), (**C**) CTRLs (white histograms) vs. active CeD (black histograms), (**D**) CTRLs (white histograms) vs. controls with early production of AGAs (AGA+CTRLs, crossed histograms). Data are shown as medians and confidence intervals of all experiments. Comparison between groups was performed by Wilcoxon Rank-Sum Test, with a *p* < 0.05 considered statistically significant and labeled with asterisks: * = *p* < 0.05, ** = *p* < 0.01.

**Table 1 ijms-24-06836-t001:** Antibody production in controls with early production of AGAs and in celiac children.

	Peak of AGA *		Peak of tTG **
Age (months)	AGA+CTRLs	CeD	CeD
6	5	0	0
12	6	0	0
18	5	1	1
24	4	4	5
30	2	5	5
36	2	8	7
42	0	2	2
48	2	3	3
54	0	3	3
60	1	3	3

Peaks of antibody production in controls with early production of AGAs (AGA+CTRLs) and celiac children (CeD). * Timing of peaks of anti-native gliadin antibodies (AGAs) have values above 10 U/mL; ** peaks of anti-tissue transglutaminase antibodies (anti-tTG) have values above 50 U/mL.

**Table 2 ijms-24-06836-t002:** Discriminant analysis based on serum cytokines and gene expression.

CTRLs vs. AGA+CTRLs		CTRLs vs. PRE-CeD		AGA+ CTRLs vs. PRE-CeD		PRE-CeD vs. CeD	
Variable	Wilks’ Lambda *	Variable	Wilks’ Lambda *	Variable	Wilks’ Lambda *	Variable	Wilks’ Lambda *
IL4 Exp	0.786	IL2	0.851	INFγ Exp	0.667	INFγ Exp	0.798
FoxP3 Exp	0.720	IL10 Exp	0.726	IL17A	0.389	TGFβ Exp	0.711
INFγ Exp	0.647	IL4	0.661	FoxP3 Exp	0.345	TNFα	0.663
IL17A	0.610	IL12	0.566	IL4 Exp	0.304	IL1β	0.541
		INFγ	0.511	IL10	0.273		
		IL17A	0.462				
		IL10	0.402				
		IL4 Exp	0.344				
		INFγ Exp	0.310				
**% Correct** **Classification**		**% Correct** **Classification**		**% Correct** **Classification**		**% Correct** **Classification**	
CTRLs	AGA+CTRLs	CTRLs	PRE-CeD	AGA+CTRLs	PRE-CeD	PRE-CeD	CeD
85%	66.7%	93.3%	88.9%	88.9%	100%	93.8%	75%

* Wilks’ lambda estimates the capacity of each variable to differentiate between the two groups being compared once the others are considered (CTRLs as controls, AGA+CTRLs as controls with AGA production, PRE-CeD as celiac patients before the diagnosis, and CeD as celiac patients after the diagnosis). In each column, the variables gene expression (name of the gene followed by Exp) and serum cytokines (only the name of the cytokine), which contribute to discriminating between the two groups, are progressively listed according to their contribution to lowering the value of Wilks’ lambda. The last line shows the percentage of children correctly predicted and classified into their original group by the discriminant equation obtained by using the combination of the above-listed variables (gene expression and serum cytokines).

**Table 3 ijms-24-06836-t003:** Children from PREVENT-CD cohort enrolled in the study and grouped based on antibody profiles.

	Definition	Status *	Age Months	AGA **	Anti-tTG **	PBMCs	Serum
CTRLs (154)	Healthy children negative for both AGAs and anti-tTG	Healthy	21.9 (16.8–26.9)	5.49 (5.29–5.69)	0.68 (0.59–0.78)	23	26
AGA+ CRTLs (28)	Healthy children producing AGAs	Healthy	21.5 (15.6–27.3)	21.66 (20.5–22.90)	0.90 (0.68–1.13)	15	16
Pre-CeD (38)	CeD at least 12 months before diagnosis	Healthy	19.3 (11.4–27.4)	18.54 (17.65–19.49)	10.98 (9.41–12.58)	16	28
CeD (38)	CeD at the time of diagnosis	Celiac	38.5 (25.9–51.1)	53.28 (44.27–64.11)	67.52 (60.48–75.38)	13	13

* Children who participated in the study were divided into 4 groups based on antibody production in the serum. ** Geometric means (antilog of the mean of log10-transformed values) and 95% confidence intervals of log10-transformed titer of antibodies are shown.

## Data Availability

The data are available on request from the corresponding author. The data are not publicly available due to privacy and ethical restrictions.

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
