# Peer review of "Antibody Profile, Gene Expression and Serum Cytokines in At-Risk Infants before the Onset of Celiac Disease"

_ijms, 2023, doi:10.3390/ijms24076836_

Round 1

Reviewer 1 Report

In this study, Auricchio et al. evaluated the presence of celiac disease-associated antibodies (AGA/D-AGA/tTG) in the serum of infants at risk for celiac disease from a longitudinal cohort. In addition, the serum concentration and gene expression of selected cytokines in peripheral blood mononuclear cells (PBMC) were assessed. In my opinion, the strength of the study includes a good number of infants from a longitudinal cohort and the panels of CD-specific antibodies tested at different months. The study's limitations include a single measure of soluble cytokines and gene expression in a small number of PBMC.

Minor improvements needed:

·        Were the infants tested for total serum IgA?

The authors need to clarify for the readers:

·        The authors describe in “Patients and study design” the criteria for diagnosis of celiac disease. Could you please include the value/titer to define “positive” for tTG and whether it was IgA or IgG?

·        It would be of interest to define if the antibodies detected are IgA or IgG for AGA, D-AGA, and tTG in the text.

·        Use either CD or CeD but not both if meaning the same. In table 2 change coeliac for celiac or use one of those.  

·        Figure 1. what does “semes” mean?

·        I would suggest the authors include descriptive figure legends. For example, in figure 1 what does each dot represent. What does the cross-line mean?

·     Figure 2 includes Tgse to represent anti-tissue transglutaminase antibodies while in the legend is refer as anti-tTG. What does DGPG mean?

·        Figure 3. Why does AGA include an “M” (AGA_M)?

·        Line 128. Figure 3 should be figure 5.

Figure 4. The authors need to include the significant p values where applied.

Author Response

In this study, Auricchio et al. evaluated the presence of celiac disease-associated antibodies (AGA/D-AGA/tTG) in the serum of infants at risk for celiac disease from a longitudinal cohort. In addition, the serum concentration and gene expression of selected cytokines in peripheral blood mononuclear cells (PBMC) were assessed. In my opinion, the strength of the study includes a good number of infants from a longitudinal cohort and the panels of CED-specific antibodies tested at different months. The study's limitations include a single measure of soluble cytokines and gene expression in a small number of PBMC. 

Minor improvements needed:

  • Were the infants tested for total serum IgA? 

 YES, according to the PREVENT-CD protocol

The authors need to clarify for the readers:

  • The authors describe in “Patients and study design” the criteria for diagnosis of celiac disease. Could you please include the value/titer to define “positive” for tTG and whether it was IgA or IgG? 

POSITIVE = ABOVE 10U/ml Only IgA are tested

  • It would be of interest to define if the antibodies detected are IgA or IgG for AGA, D-AGA, and tTG in the text. 
  • Use either CED or CeD but not both if meaning the same. In table 2 change coeliac for celiac or use one of those.  

OK

  • Figure 1. what does “semes” mean?

DELETED   semes = semesters 6 months

  • I would suggest the authors include descriptive figure legends. For example, in figure 1 what does each dot represent. What does the cross-line mean? 

DATA FROM FIG1,2,3 ARE NOW LISTED IN TABLE 1  and clear legends are available

  • Figure 2 includes Tgse to represent anti-tissue transglutaminase antibodies while in the legend is refer as anti-tTG. What does DGPG mean? 

Now all tTG for Anti Transglutaminase and Deamidated Gliadin peptide (IgA and IgG) for DGPG-A

  • Figure 3. Why does AGA include an “M” (AGA_M)?

Error, sorry

  • Line 128. Figure 3 should be figure 5. 

OK

Figure 4. The authors need to include the significant p values where applied.

OK

Reviewer 2 Report

The manuscript entitled „ ANTIBODIES PROFILE, GENE EXPRESSION AND SERUM CYTOKINES IN AT-RISK INFANTS BEFORE THE ONSET OF CELIAC DISEASE” presents interesting issue, but some problems should be corrected.

Major:

1.       Authors should present a proper literature background for their study. The current version of the manuscript includes only 13 references, which con not be perceived as a proper literature background.

2.       Authors should properly prepare their study, as now it is shabbily prepared (e.g. for celiac disease they use various abbreviations: CD, or CeD)

3.       Authors should refer relevant references, associated with the information presented (e.g. for the sentence about serum antibodies against native gliadin and deamidated gliadin, and against tissue transglutaminase, they refer their own reference entitled “Randomized feeding intervention in infants at high risk for celiac disease” (not associated with the presented information)

4.       Authors should correct their manuscript as current version is shabbily prepared (e.g. words not being in English – “semes”?)

5.       The figures/ tables should be stand-alone ones – be able to be understand without reading the manuscript, so all the abbreviations should be explained in footnotes.

Abstract:

Instead of what was done („we explored…”) Authors should formulate what was the aim of the study (e.g. „The aim of the study was…”).

Authors should clearly describe the studied cohort.

Authors should present specific numeric results accompanied by the results of the statistical analysis (p-Values).

Keywords:

More keywords, especially more specific for the conducted study, should be presented.

Introduction:

Each presented information should include specific reference, while in the current version of the manuscript there are whole paragraphs without a single reference (lines 56-64).

Results:

Instead of Figures Authors should present rather tables as their figures are extremely hard to follow (especially Figure 3)

Based on SD values presented it seems that Authors did not verify normality of distribution.

Authors should verify normality of distribution and only for parametric data they should present mean and SD, while for non-parametric they should present median, min and max values.

Authors should use statistical tests based on the distributions observed.

Discussion:

This section should be deepened - Authors should: (1) compare gathered data with the results by other authors, (2) formulate implications of the results of their study and studies by other authors, (3) formulate the future areas which should be studied.

Conclusions:

Conclusions should be clearly formulated.

Materials and methods:

Authors should describe the studied sample (cohort) – how was it obtained, from what region/ which cities, etc.

It seems that Authors did not verify normality of distribution.

Authors should verify normality of distribution and only for parametric data they should present mean and SD, while for non-parametric they should present median, min and max values.

Authors should use statistical tests based on the distributions observed.

References:

Author should include adequate references, while self-citations should be avoided, as they are not adequate (8 of 13 references – over 60% are the own references by Authors) – based on such proportions it may be stated that Authors are not familiar with the current literature, but only with their own studies

Author Response

  1. Authors should present a proper literature background for their study. The current version of the manuscript includes only 13 references, which con not be perceived as a proper literature background.

NOW DONE, background included, literature expended, thanks for the suggestion

  1. Authors should properly prepare their study, as now it is shabbily prepared (e.g. for celiac disease they use various abbreviations: CED, or CeD)

OK

  1. Authors should refer relevant references, associated with the information presented (e.g. for the sentence about serum antibodies against native gliadin and deamidated gliadin, and against tissue transglutaminase, they refer their own reference entitled “Randomized feeding intervention in infants at high risk for celiac disease” (not associated with the presented information) 

NOW corrected either in the Introduction as well as in the Discussion

  1. Authors should correct their manuscript as current version is shabbily prepared (e.g. words not being in English – “semes”?)

DONE, all revise semes for semester

  1. The figures/ tables should be stand-alone ones – be able to be understand without reading the manuscript, so all the abbreviations should be explained in footnotes.

Figures now simplified with clear legends to stand alone

Abstract:

Instead of what was done („we explored…”) Authors should formulate what was the aim of the study (e.g. „The aim of the study was…”).

Now revised

Authors should clearly describe the studied cohort.

DONE

Authors should present specific numeric results accompanied by the results of the statistical analysis (p-Values).

 Now in Table instead of relatively confusing graphs

Keywords:

More keywords, especially more specific for the conducted study, should be presented.

OK

Introduction:

Each presented information should include specific reference, while in the current version of the manuscript there are whole paragraphs without a single reference (lines 56-64).

Now done

Results:

Instead of Figures Authors should present rather tables as their figures are extremely hard to follow (especially Figure 3)

NOW DATA OF FIG 1,2 3 ARE IN TABLE  1

Based on SD values presented it seems that Authors did not verify normality of distribution.

DONE

Authors should verify normality of distribution and only for parametric data they should present mean and SD, while for non-parametric they should present median, min and max values.

Authors should use statistical tests based on the distributions observed.

Since the distribution of antibodies does not respect normality, a 10Log transformation was applied and geometric means, with 95% Confidence Intervals now shown

Discussion:

This section should be deepened - Authors should: (1) compare gathered data with the results by other authors, (2) formulate implications of the results of their study and studies by other authors, (3) formulate the future areas which should be studied.

Now all completely revised according to suggestions

Conclusions:

Conclusions should be clearly formulated.

OK

Materials and methods:

Authors should describe the studied sample (cohort) – how was it obtained, from what region/ which cities, etc.

It seems that Authors did not verify normality of distribution.

Authors should verify normality of distribution and only for parametric data they should present mean and SD, while for non-parametric they should present median, min and max values.

Authors should use statistical tests based on the distributions observed.

We feel sorry for the attempt to show raw numbers of antibodies in table 2. We are aware that all antibodies do not have a normal distribution (skewness > 4.0 as well as Curtosis > 10) . Hence we should not have shown these means in table 2. When median are used, the range (min/max) of values is so large to limit the interpretation of data. (The max values is > 100 for Anti-Transglutaminase, for example). These antibodies have a log distribution: when decimal log is applied Skewnes and Curtosis fall below 1 and they distributed normally. Now in table 2 we show the anti-log of geometric means (means of decimal log) with the correspondent 95% Confidence Intervals. (Armitage P. Statistical Methods in Medical Research. 1971. Blackwell Scientific Publications).

References:

Author should include adequate references, while self-citations should be avoided, as they are not adequate (8 of 13 references – over 60% are the own references by Authors) – based on such proportions it may be stated that Authors are not familiar with the current literature, but only with their own studies OK

Now all enriched

Round 2

Reviewer 2 Report

The manuscript entitled „ ANTIBODIES PROFILE, GENE EXPRESSION AND SERUM CYTOKINES IN AT-RISK INFANTS BEFORE THE ONSET OF CELIAC DISEASE” presents interesting issue, but some problems should be corrected.

Major:

1.       Authors should present a proper literature background for their study. The current version of the manuscript includes only 17 references, which con not be perceived as a proper literature background.

2.       Authors should properly prepare their study, as now it is shabbily prepared (e.g. Wilks’ Lambda in their tables is referred as “Wilks’” – it is not a common reference)

3.       Authors should refer relevant references, associated with the information presented (e.g. for the sentence presenting definition of coeliac disease, they refer 2 positions – entitled “Identification and Analysis of Multivalent Proteolytically Resistant Peptides from Gluten” and “Nomenclature and listing of celiac disease relevant gluten T-cell epitopes restricted by HLA-DQ molecules” (none of them is directly associated with the presented information being not associated with coeliac disease itself)

4.       Authors should correct their manuscript as current version is shabbily prepared (e.g. title of table formulated as “shows the peaks of production of antibodies in healthy controls (second column) and in CeD. Peaks Of Antibodies Production In Aga+Ctrls And Ced” is improper – it is not a title but a description)

5.       The figures/ tables should be stand-alone ones – be able to be understand without reading the manuscript, so all the abbreviations should be explained in footnotes.

Abstract:

Authors should clearly describe the studied cohort.

Authors should present specific numeric results accompanied by the results of the statistical analysis (p-Values).

Keywords:

More keywords, especially more specific for the conducted study, should be presented.

Introduction:

Each presented information should include specific reference, while in the current version of the manuscript there are whole paragraphs without a single reference (lines 62-69).

Results:

Instead of Figures Authors should present rather tables as their figures are extremely hard to follow (especially Figure 3)

Discussion:

Discussion must be based on the literature (specific references) – Authors can not conduct discussion without referring specific information from literature

Conclusions:

Conclusions should be clearly formulated.

Materials and methods:

Authors should describe the studied sample (cohort) – how was it obtained, from what region/ which cities, etc.

References:

Author should include adequate references, while self-citations should be avoided, as they are not adequate (7 of 17 references – over 41% are the own references by Authors) – based on such proportions it may be stated that Authors are not familiar with the current literature, but only with their own studies

Author Response

Review Report Manuscript ID: ijms-1839481

The manuscript entitled „ ANTIBODIES PROFILE, GENE EXPRESSION AND SERUM CYTOKINES IN AT-RISK INFANTS BEFORE THE ONSET OF CELIAC DISEASE” presents interesting issue, but some problems should be corrected.

Major:

  1. Authors should present a proper literature background for their study. The current version of the manuscript includes only 17 references, which con not be perceived as a proper literature background.

A: We have enriched the manuscript with new and appropriate references.

  1. Authors should properly prepare their study, as now it is shabbily prepared (e.g. Wilks’ Lambda in their tables is referred as “Wilks’” – it is not a common reference)

A: We have noticed the inaccuracy of some sentences, and we have improved our manuscript in form and content.

  1. Authors should refer relevant references, associated with the information presented (e.g. for the sentence presenting definition of coeliac disease, they refer 2 positions – entitled “Identification and Analysis of Multivalent Proteolytically Resistant Peptides from Gluten” and “Nomenclature and listing of celiac disease relevant gluten T-cell epitopes restricted by HLA-DQ molecules” (none of them is directly associated with the presented information being not associated with coeliac disease itself)

A: We have corrected and added some references, as requested.

  1. Authors should correct their manuscript as current version is shabbily prepared (e.g. title of table formulated as “shows the peaks of production of antibodies in healthy controls (second column) and in CeD. Peaks Of Antibodies Production In Aga+Ctrls And Ced” is improper – it is not a title but a description)

A: We have corrected some inaccuracies in the form of our manuscript.

  1. The figures/ tables should be stand-alone ones – be able to be understand without reading the manuscript, so all the abbreviations should be explained in footnotes.

A: We thank the reviewer for noting the lack of immediate clarity of the previous figures and tables, these latters have now been better explained.

Abstract:

Authors should clearly describe the studied cohort.

Authors should present specific numeric results accompanied by the results of the statistical analysis (p-Values).

A: We have added the statistical differences, when possible, in the abstract.

Keywords:

More keywords, especially more specific for the conducted study, should be presented.

A: We have modified the keywords by inserting some more specific to our study.

Introduction:

Each presented information should include specific reference, while in the current version of the manuscript there are whole paragraphs without a single reference (lines 62-69).

A: As already mentioned, we added some references in all the text of the manuscript.

Results:

Instead of Figures Authors should present rather tables as their figures are extremely hard to follow (especially Figure 3)

A: In the previous revised version of the manuscript, we have already changed the Figure 3 into a Table, since we agreed with the Reviewer on the difficulty of reading it.

Discussion:

Discussion must be based on the literature (specific references) – Authors can not conduct discussion without referring specific information from literature

A: We have added references also in this section.

Conclusions:

Conclusions should be clearly formulated.

A: We have tried to make our discussion clearer.

Materials and methods:

Authors should describe the studied sample (cohort) – how was it obtained, from what region/ which cities, etc.

A: We have better described the prospective cohort enrolled in our study in the current version of the manuscript.

References:

Author should include adequate references, while self-citations should be avoided, as they are not adequate (7 of 17 references – over 41% are the own references by Authors) – based on such proportions it may be stated that Authors are not familiar with the current literature, but only with their own studies

A: As previously mentioned, we added and modified several references, in the current manuscript there are 31.

Round 3

Reviewer 2 Report

The manuscript entitled „ ANTIBODIES PROFILE, GENE EXPRESSION AND SERUM CYTOKINES IN AT-RISK INFANTS BEFORE THE ONSET OF CELIAC DISEASE” presents interesting issue, but some problems should be corrected.

Major:

1.       Authors should present a proper literature background for their study. The current version of the manuscript includes only 33 references, which can not be perceived as a proper literature background.

2.       Authors should correct their manuscript as current version is shabbily prepared (e.g. title of table formulated as “Discriminant analysis of biomarkers” is improper – it is not a title but a description)

3.       The figures/ tables should be stand-alone ones – be able to be understand without reading the manuscript, so all the abbreviations should be explained in footnotes.

Results:

Instead of Figures Authors should present rather tables as their figures are extremely hard to follow (Figure 1, Figure 2)

Conclusions:

Conclusions should be clearly formulated.

Materials and methods:

Authors should describe the studied sample (cohort) – how was it obtained, from what region/ which cities, etc.

References:

Author should include adequate references, while self-citations should be avoided, as they are not adequate (12 of 32 references – over 37% are the own references by Authors) – based on such proportions it may be stated that Authors are not familiar with the current literature, but only with their own studies

Author Response

Major:

  1. Authors should present a proper literature background for their study. The current version of the manuscript includes only 33 references, which cannot be perceived as a proper literature background.

We thank the reviewer for suggesting us to enrich the bibliography. To our knowledge, we believe to have cited the most updated studies in this extremely innovative field of investigation.

In any case, we have further revised the literature reaching 35 references (in the previous version there were 31), by adding 4 new manuscripts collected with a new search on PubMed with these keywords: “antibodies and celiac disease and risk”:

  • reference number 30 (Simell S et al, American Journal of Gastroenterology, 2007, 102(9):2026-35)
  • reference number 31 (Hoffenberg EJ et al, Journal of Pediatrics, 2000, 137(3):356-60)
  • reference number 32 (Not T et al, Scandinavian Journal of Gastroenterology, 1998, 33(5):494-8)
  • reference number 33 (Lähdeaho ML et al, International Archives of Allergy and Immunology, 1993;101(3):272-6)

All of them added in the paragraph 3. Discussion, at page 9 in line 274.

  1. Authors should correct their manuscript as current version is shabbily prepared (e.g. title of table formulated as “Discriminant analysis of biomarkers” is improper – it is not a title but a description).

We have done our best to clarify the title of Table 2: “Discriminant analysis based on serum cytokines and gene expression”, instead of “Discriminant analysis of biomarkers”.

It is required that the tables and figures are self-explanatory, consequently their title should also contain useful information to make the data shown usable without needing to read the manuscript. Indeed, we will greatly appreciate if this Reviewer could suggest us a proper title for Table 2.

  1. The figures/ tables should be stand-alone ones – be able to be understand without reading the manuscript, so all the abbreviations should be explained in footnotes.

Thanks for noticing, we corrected and clarified the abbreviations in the following tables and figures in the footnotes:

  • Table 1 page 4 lines 94-95,
  • Figure 2 page 6 lines 155, 157-158.
  • Table 2 page 7, lines 182-183.

Results:

Instead of Figures Authors should present rather tables as their figures are extremely hard to follow (Figure 1, Figure 2).

We thank the reviewer for this suggestion, we have already taken the reviewer’s advice into account and replaced the figure with a table whenever possible (previous figure 1 A, B, C has become table 1). For the Figures 1 and 2, as we do not believe the data can be more understandable if presented in Tables but thanking in account the reviewer’s suggestion, we have tried to better clarify them.

In particular, in Figure 1 we have chosen different colours and symbols for each children group and separated with a line each evaluated serum cytokine.

In the Figure 2, we have modified the ordinate axis by changing RQ to Relative Quantities and the abscissa axis adding a curly brace to better divide the expression of selected cytokine genes.

Conclusions:

Conclusions should be clearly formulated.

We have tried to formulate the conclusions more clearly by separating them from the discussion and creating a new paragraph: 4. Conclusions, page 9 from line 277, and adding the following sentence at page 9 in the lines 277-283:

“In conclusion, we demonstrated an early production of gliadin IgA antibodies (AGA) in a subset of at risk infants for CeD, but this is not predictive of the disease. On the contrary, the early production of AGA seems to protect them. In contrast, in children who later develop CeD, there is a simultaneous production of AGA, anti-DGP and anti-tTG antibodies. Furthermore, in those who develop the disease, changes in the gene expression and cytokine pattern precede the appearance of anti-tTG antibodies.”

Materials and methods:

Authors should describe the studied sample (cohort) – how was it obtained, from what region/ which cities, etc.

The study cohort was enrolled in Naples, Italy, as better clarified now in the text, in the paragraph 5.1. Patients and study design of Materials and Methods, page 9, lines 288-293:

An Italian cohort of 220 HLA DQ2/DQ8+ infants from families with a celiac case was followed from birth to 6 years of age. In particular, about 90% of the enrolled children came from the metropolitan area of Naples, while the remaining 10% from other territorial areas of the Campania region. Children were followed at the Department of Translational Medical Sciences, Section of Pediatrics, Federico II, University of Naples”.

References:

Author should include adequate references, while self-citations should be avoided, as they are not adequate (12 of 32 references – over 37% are the own references by Authors) – based on such proportions it may be stated that Authors are not familiar with the current literature, but only with their own studies

As reported above, we have integrated references from 13 of the first submission to 35 of this last,

reducing self-citations from 60% to 25%. Notwithstanding, our research group has contributed, like few other authors, to the study of risk factors associated with the development of celiac disease in children at very early age. Unfortunately, as answered to the reviewer in the previous comment, we had difficulty citing other contributions, given the extremely innovative topic, but nevertheless we have tried to enrich the bibliography by adding the aforementioned references.

Round 4

Reviewer 2 Report

The manuscript entitled „ ANTIBODIES PROFILE, GENE EXPRESSION AND SERUM CYTOKINES IN AT-RISK INFANTS BEFORE THE ONSET OF CELIAC DISEASE” presents interesting issue, but some problems should be corrected.

Major:

Authors should present a proper literature background for their study. The current version of the manuscript includes only 35 references, which can not be perceived as a proper literature background, especially if a number of them are old and very old publications (13 references from 35 cited, being almost 40%, are published until 2010, including number of publications published in 1990s). Moreover, self-citations should be avoided, as they are not adequate (12 of 35 references – over 34% are the own references by Authors) – based on such proportions it may be stated that Authors are not familiar with the current literature, but only with their own studies, as they referred either very old studies or their own studies.

Results:

Instead of Figures Authors should present rather tables as their figures are extremely hard to follow (Figure 1, Figure 2)

Materials and methods:

Authors should describe in details the studied sample (cohort) – how was it obtained, from which cities, etc.

References:

Author should include adequate references, while self-citations should be avoided, as they are not adequate (12 of 35 references – over 34% are the own references by Authors) – based on such proportions it may be stated that Authors are not familiar with the current literature, but only with their own studies

Author Response

(The authors gave the same response as above.)
